# The Distal and Local Volcanic Ash in the Late Pleistocene Sediments of the Termination I Interval at the Reykjanes Ridge, North Atlantic, Based on the Study of the Core AMK-340

**Alexander Matul** [1,*] , **Irina F. Gablina** [2], **Tatyana A. Khusid** [1], **Natalya V. Libina** [1] and **Antonina I. Mikhailova** [2]

1   Shirshov Institute of Oceanology, Russian Academy of Sciences, 117997 Moscow, Russia
2   Geological Institute, Russian Academy of Sciences, 119107 Moscow, Russia
*   Correspondence: amatul@mail.ru

**Abstract:** We made the geochemical analysis of the volcanic material from the sediment core AMK-340 (the Russian research vessel "Akademik Mstislav Keldysh" station 340), the central zone of the Reykjanes Ridge. Two ash-bearing sediment units within the interval of the Termination I can be detected. They correlate with the Ash Zone I in the North Atlantic Late Quaternary sediments having an age of 12,170–12,840 years within the Younger Dryas cold chronozone and 13,600–14,540 years within the Bølling–Allerød warm chronozone. The ash of the Younger Dryas unit is presented mostly by the mafic and persilicic material originated from the Icelandic volcanoes. One sediment sample from this unit contained Vedde Ash material. The ash of the Bølling–Allerød unit is presented mostly by the mafic shards which are related to the basalts of the rift zone on the Reykjanes Ridge, having presumably local origin. Possible detection of Vedde Ash could help to specify the timing of the previously reconstructed paleoceanographic changes for the Termination I in the point of the study: significant warming in the area might have occurred as early as 300 years before the end of the conventional Younger Dryas cold chronozone.

**Keywords:** tephra in marine sediments; Ash Zone I in North Atlantic; tephrochronology of Termination I

## 1. Introduction

Tephrochronology is a widely used tool for dating and correlating the marine and terrestrial sediment sequences, especially within the Quaternary [1]. Recent detailed mineralogical and geochemical studies of volcanic material revealed a high-resolution Late Pleistocene and Holocene chronostratigraphy for the North Atlantic [2–4]. Icelandic volcanoes are the major source of the ash in the marine sediments of the Nordic Seas and North Atlantic [5]. Extensive studies of the Icelandic soil, lake, and shelf sediments documented >150 tephra layers within the late glacial and Holocene time [6]. Thornalley et al. [7] detected numerous ash-bearing marine sediment layers south of Iceland within the last deglacial and Holocene time. Such data help to refine the regional and local sediment stratigraphy and synchronize the paleoclimatic archives between the distal oceanic and land regions. Numerous studies of the Late Pleistocene tephra layers in the North Atlantic domain exhibited a complicated mineral/geochemical composition of the ash material accumulated in the marine sediments during the Termination I between approximately 18 and 11 ka (e.g., [8–10]). Different processes influence the deposition of the specific ash layers: single eruption or closely-timed chain of eruptions in one locality,

dispersal by the atmospheric and oceanic flows, iceberg and sea-ice rafting, and reworking by the bottom currents and bioturbation [11].

Our study's aim is to get additional information on the occurrence and composition of the tephra in the North Atlantic sediments at the transition from the last glacial period to the Holocene. The position of the sediment core AMK-340 (the Russian research vessel "Akademik Mstislav Keldysh" station 340) is in the central zone of the Reykjanes Ridge with a local source of the eruptive material. Therefore, we tried to recognize the different sources of the volcanic ash, local or distal ones. Geochemical analysis of the volcanic shards using scanning electron microscopy will help to reveal the specific, well-known tephra layers like Vedde Ash which can be a good marker for a refinement of the core stratigraphy and chronology of the local paleoceanographic changes.

## 2. Materials and Methods

The sediment core AMK-340 was obtained during the 4th cruise of the Russian research vessel "Akademik Mstislav Keldysh" [12] in the central part of the Reykjanes Ridge, North Atlantic south of Iceland (Figure 1): 58°30.6′ N, 31°31.2′ W, water depth of 1689 m, core length 387 cm.

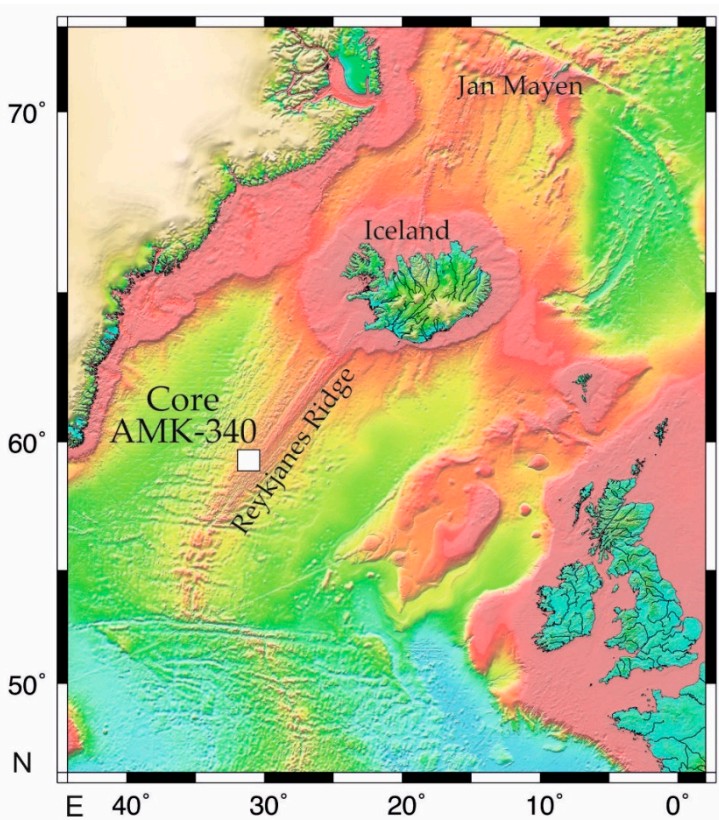

**Figure 1.** The geographic position of the sediment core AMK-340 (the Russian research vessel "Akademik Mstislav Keldysh" station 340). Source of the map is (https://topex.ucsd.edu/marine_topo/jpg_images/topo4.jpg).

Figure 2 presents some general litho- and biostratigraphic and chronological data on the core AMK-340. More data are presented in [13–15]. The core unit of 0–307 cm is composed of the white pelitic calcareous to weakly calcareous (foraminiferal-coccolith) oozes with a $CaCO_3$ content from 10–25% to 40–50%. In the lower core unit of 307–387 cm, the sediment color becomes mainly grey with thin alternations of greenish-grey, yellowish-grey, and dark, almost black, bands. This unit is enriched in the diatoms (sometimes up to 10–30%). The $CaCO_3$ content varies there between 5.5% and 20% [12]. A visual lithological description of the core exhibited no signs of the volcanic material.

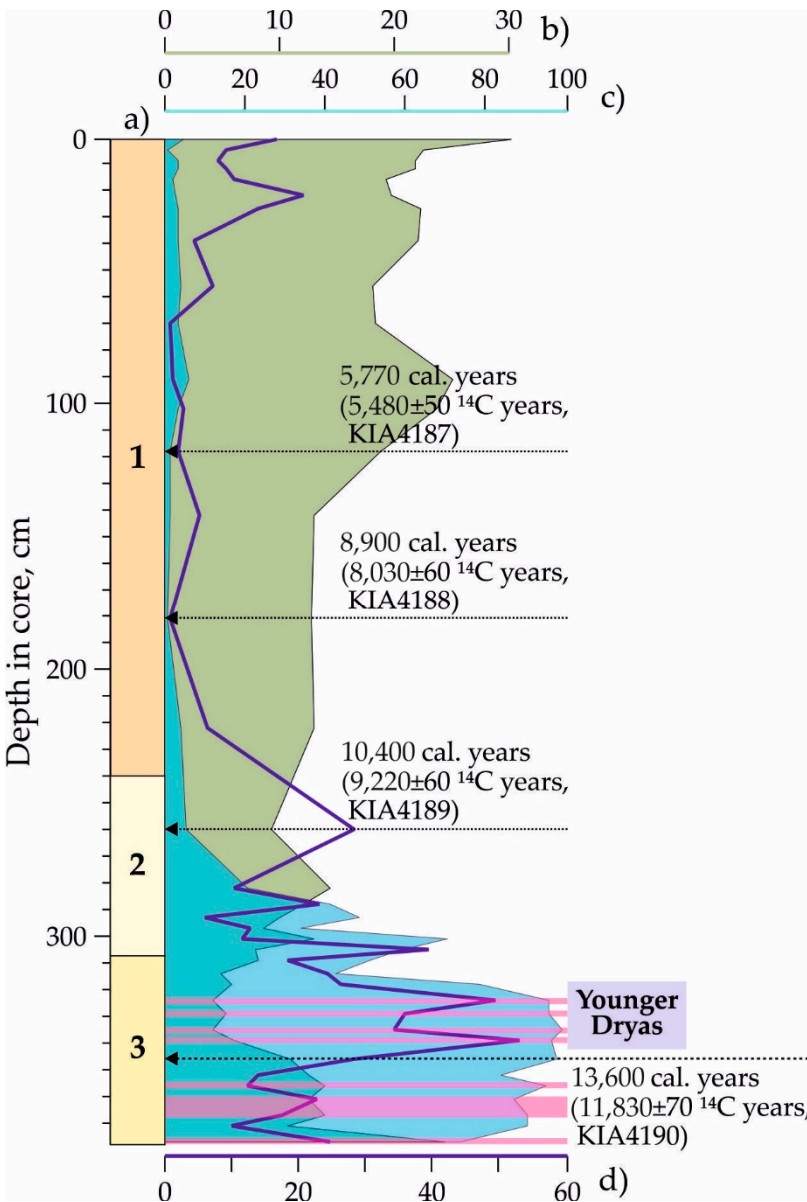

**Figure 2.** Lithostratigraphy of the AMK-340. Lithological description (**a**): number 1 is pelitic calcareous oozes with a $CaCO_3$ content of >40–50%, number 2 is pelitic weakly calcareous muds with a $CaCO_3$ content of 10–25%, and number 3 is silty-pelitic muds with a $CaCO_3$ content between 5.5% and 20% and an admixture of biogenic $SiO_2$ of up to 10–30% [12]. The green-colored graph (**b**) is a distribution of the radiolarian species *Lithelius spiralis* group as a marker of the temperate (non-glacial) conditions in the North Atlantic [13]. The blue-colored graph (**c**) is a distribution of the polar planktic foraminiferal species *Neogloboquadrina pachyderma* sinistral [13] as a glacial marker. The dark-blue line (**d**) is a distribution of the mineral grains in the sediment fraction of >100 μm [13] as a marker of the ice-rafted material. Pink bars are the samples where the ash material was analyzed. Black dotted arrows with captions show positions of the AMS (Accelerator Mass Spectrometry) [14]C-dates. Violet rectangle indicates the Younger Dryas unit.

The linear interpolation between four AMS [14]C-datings [13] (Figure 2) helped to develop an age model of the core AMK-340, taking into account litho- and biostratigraphy [14,15].

Calibration of the 14C-datings in the calendar age data was made in the CALIB 7.1.0 computer software using the MARINE13 calibration scale with the standard reservoir correction of 405 years and its regional deviation of 85 ± 79 years [16]. The core spans the time interval of the last approximately

14,500 years or the Termination I and Holocene. We assign the core unit of 315–340 cm the Younger Dryas interval based, in addition to the radiocarbon dating, on the litho- and biostratigraphic information [13,15]. As in many other northern North Atlantic sediment cores, an abundance peak of the polar planktic foraminiferal species *Neogloboquadrina pachyderma* sinistral (left-coiled shells) of up to >99% together with an occurrence of the Heinrich 0 layer (maximum of ice-rafted material accumulation—see a mineral grains distribution in Figure 2) marks the Younger Dryas cooling event. In the same unit, the radiolarian species *Amphimelissa setosa*, which is now typical for the cold-water environments of the Nordic Seas and the Arctic, has a very noticeable increase in abundance; overall, the radiolarian assemblages were subarctic there. Rich benthic foraminiferal microfauna, including the specific bioproductivity-indicating species, appeared in the core just above this unit. A possible detection of Vedde Ash with the age of 12,170 years [3,17,18] in the sample 323–325 cm (see Discussion Section) allows us to make a more accurate age model of the sediment core AMK-340 for the time interval of the Termination I (sharp warming with the climatic fluctuations between the Last Glacial Maximum and Holocene). From new calculations regarding possible Vedde Ash detection, the lower time limit of the core AMK-340 could be 14,540 years B.P. The core units of 323–340 and 355–378 cm could have the presumable age of 12,170–12,840 and 13,600–14,540 years, respectively.

During the micropaleontological analysis of the core AMK-340 sediments under the stereomicroscope (foraminiferal study of the sediment fraction of >100 μm) and transmitted light microscope (radiolarian study of the sediment fraction of >50 μm) (Figure 3), we could recognize a remarkable admixture of the eruptive shards in the two core units (Figure 1), 323–340 cm (four samples: 323–325, 328–330, 334–336, and 338–340 cm) and 355–378 cm (four samples: 355–357, 360–368, 370–372, and 376–378 cm). In all these samples, the content of the eruptive material was >10–15% from the whole number of grains in the sediment fraction >100 μm. Volcanic shards in other samples of the core within the interval of 0–323 cm were absent or occasional. From these samples, eruptive shards in the fraction >100 μm were picked out for the subsequent studies of their chemical composition. All in all, data on 16 shards in the natural state, 24 shards from the core unit of 323–340 cm, and 16 shards from the core unit of 355–378 cm in the polished thin sections are presented (Table 1).

Two instruments examined the chemical composition of the shards. The first one is a scanning electron microscope CamScan MV2300 with the energy dispersive analysis system (EDS) INCA at the Geological Institute of the Russian Academy of Sciences, Moscow, Russia, (gold-sprayed shards in the natural state). The selected locality of measurements was 1 μm, and the voltage was 20 kV. The microanalyzer was equipped with a mylar input window of 0.5 μm in caliber which allowed to analyze light elements including oxygen and carbon. The reference samples for the quantitative analysis were obtained from the standard mineral collection provided by Oxford Instruments. From this collection, an analytical program tried to find a number of reference samples which were chemically close to the analyzed samples. The analytical accuracy was as high as 0.15%. Further, the scanning electron microscope VEGA3 TESCAN with EDS (Electron Dispersive Spectrometry) X-Max was used at the Institute of Oceanology of the Russian Academy of Sciences, Moscow, Russia (carbon-sprayed shards in the polished thin sections). The diameter of the electron beam was 1 μm, and the accelerating voltage was 20 kV. The concentrations of elements were detected using a correction on the absorption and dispersion of the X-rays with the help of the INCA standard software. The analytical accuracy was as high as 0.2%. Table S1 presents the normalized (along with some raw) data of the measurements.

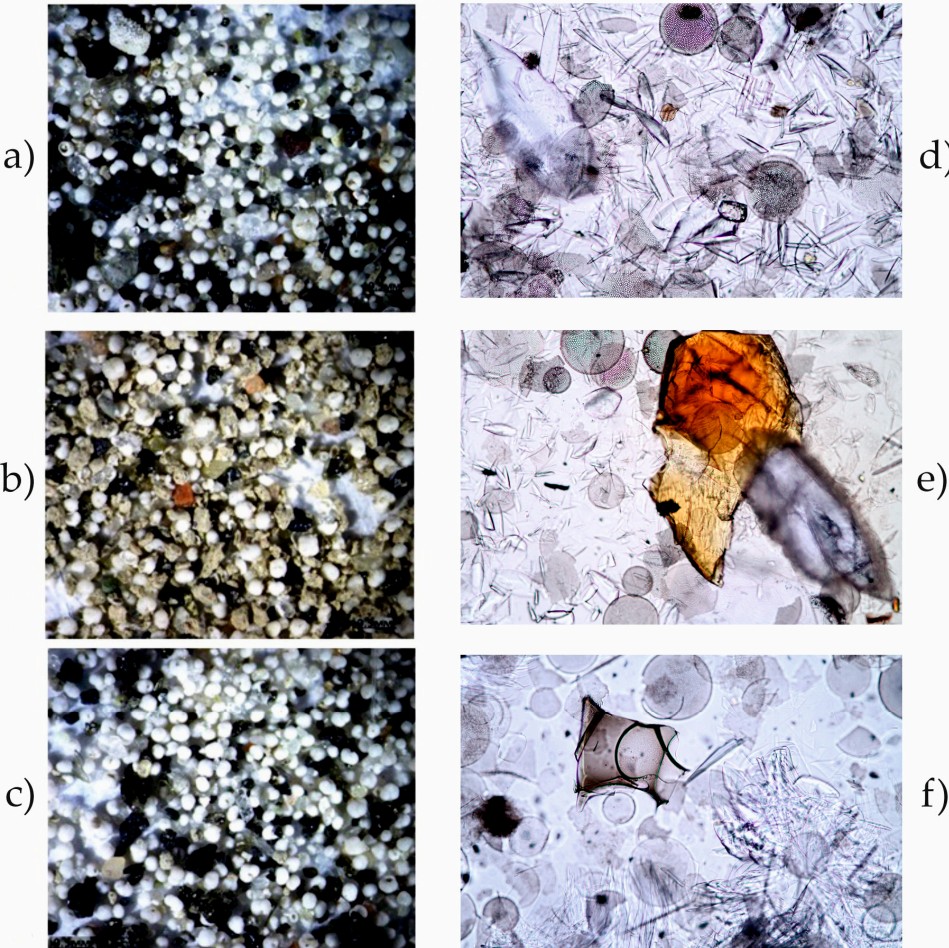

**Figure 3.** Microphotographs of the sediment samples of the core AMK-340. Left panel present photos of the sediment fraction >0.1 mm under the stereomicroscope: (**a**) core sample of 323–325 cm, (**b**) sample of 328–330 cm, (**c**) sample of 338–340 cm. Right panel present photographs of the sediment fraction >0.05 mm under the transmitted light microscope: (**d**) core sample of 323–325 cm, (**e**) sample of 328–330 cm, (**f**) sample of 338–340 cm. The regular microphotographs of the sediment fractions were obtained with the stereomicroscope Zeiss Stemi 508 equipped with the camera AxioCam Icc5 (left panel), and the transmitted light microscope used was the Zeiss Axio Lab.A1 powered by the Canon EOS 1100D camera (right panel).

**Table 1.** The number of shards and measurements in the studied samples of the core AMK-340.

| Core Sample, cm | Shards in Total | Shards in Polished Thin Sections | Measurements in Total | Measurements in Polished Thin Sections |
|---|---|---|---|---|
| 323–325 | 9 | 3 | 23 | 12 |
| 328–330 | 9 | 5 | 20 | 14 |
| 334–336 | 13 | 7 | 23 | 13 |
| 338–340 | 9 | 9 | 29 | 29 |
| **Core Unit 323–340** | **40** | **24** | **95** | **66** |
| 355–357 | 3 | 3 | 8 | 8 |
| 360–368 | 5 | 5 | 15 | 15 |
| 370–372 | 3 | 3 | 10 | 10 |
| 376–378 | 5 | 5 | 13 | 13 |
| **Core Unit 355–378** | **16** | **16** | **46** | **46** |
| **In Total** | **56** | **40** | **141** | **112** |

## 3. Results

### 3.1. Morphological Types of the Eruptive Material in the Studied Sediment Samples

Angular fragments of the pumiceous basalts and basaltic andesites were black and sometimes greenish, aphanitic (microlithic) and porous, with cavities filled with the light volcanic ash and sometimes with the fragmented diatom frustules (Figure 4).

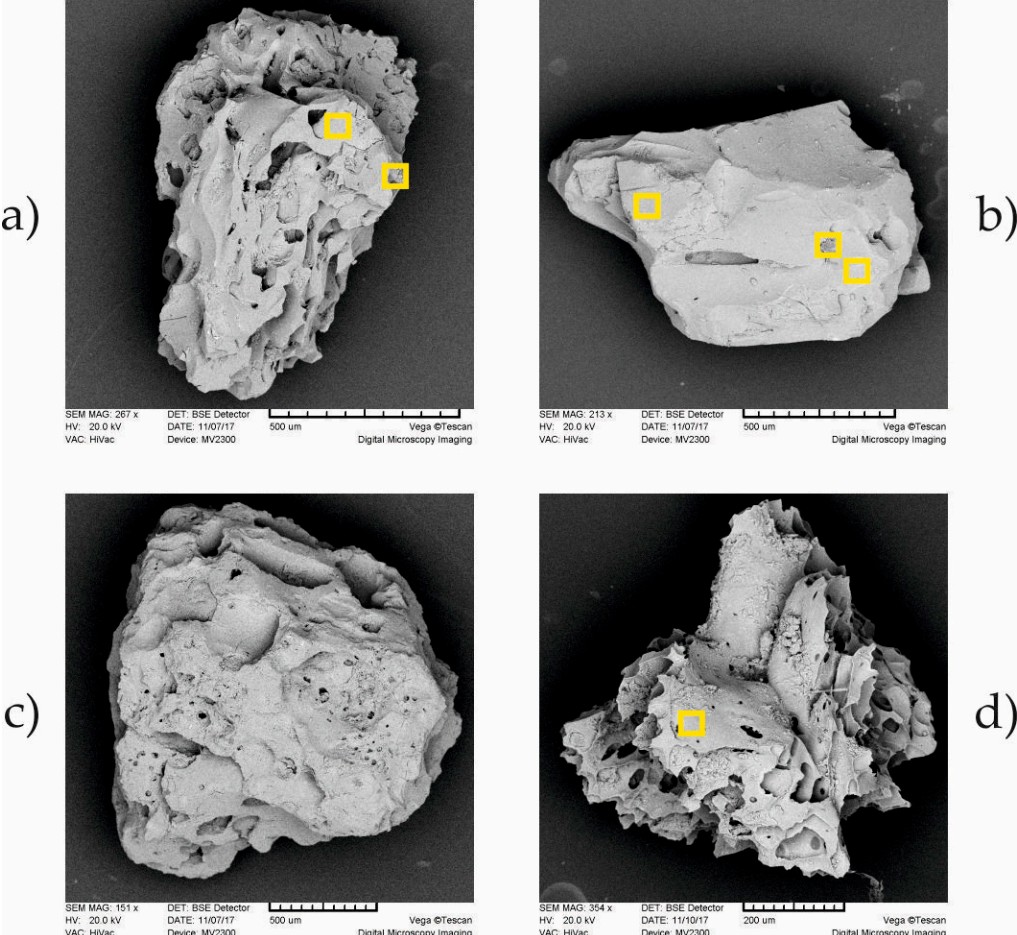

**Figure 4.** Scanning electron microphotographs of the basalt and andesitic basalt ash shards from the core unit of 323–340 cm: (**a**) vesicular black basalt glass with a $SiO_2$ content of 49.10–57.15% in the sample of 323–325 cm, (**b**) massive black semi-transparent glass with a $SiO_2$ content of 49.45% in the sample of 323–325 cm, (**c**) vesicular black shard ($SiO_2$ content of 50.22%) with cavities filled with the andesitic ($SiO_2$ content of 55.79%) and persilicic ($SiO_2$ content of 63.89–65.97%) dust in the sample of 323–325 cm, (**d**) highly vesicular dark-green semi-transparent andesitic basalt glass with a $SiO_2$ content of 52.52% in the sample of 334–336 cm. Yellow-outlined rectangles show points of measurements.

Angular transparent/subtransparent fragments of the persilicic glass were olive and bottle-green, elongate (columnar) with rough scratching on the surface parallel to the elongation or having an irregular shape, and aphanitic (Figure 5).

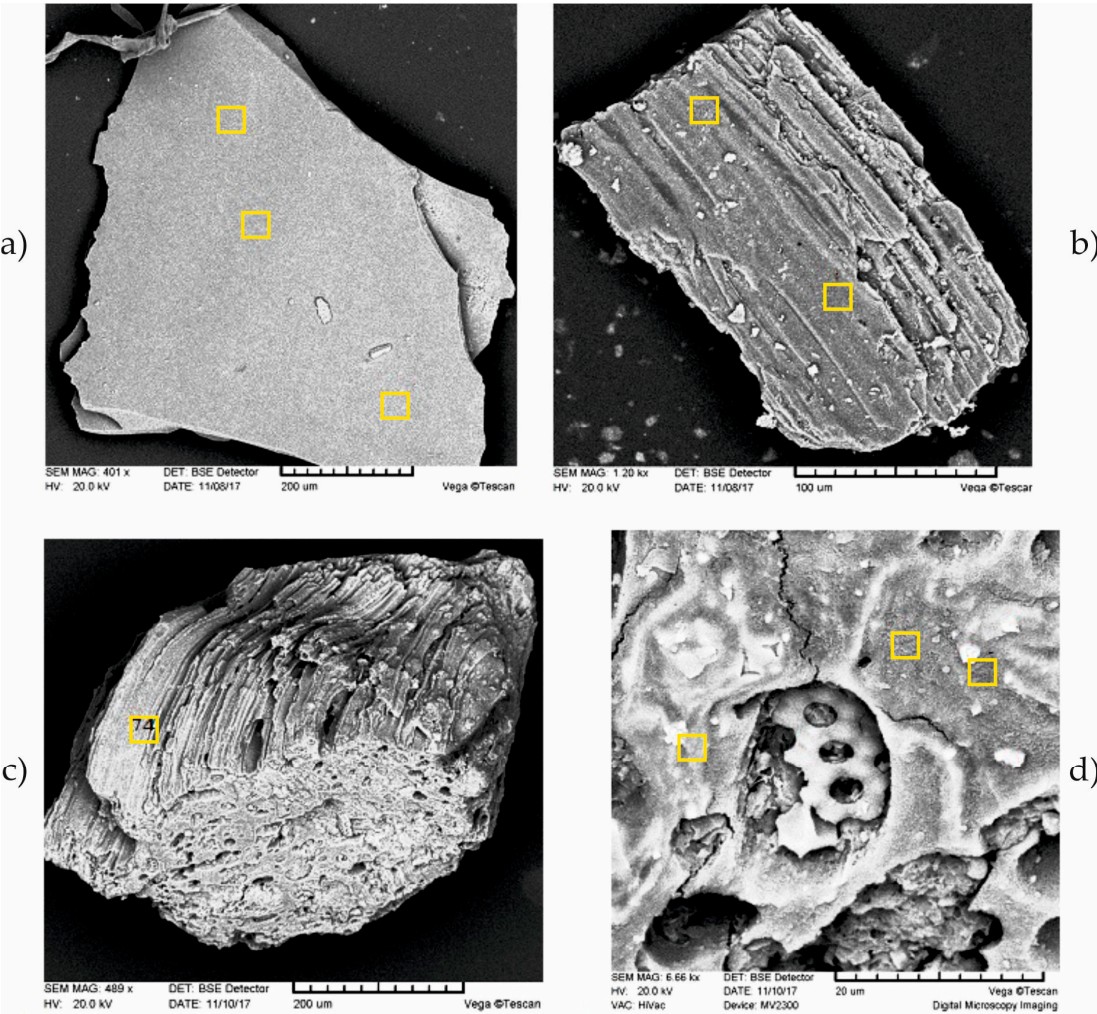

**Figure 5.** Scanning electron microphotographs of the persilicic ash shards from the core unit of 323–340 cm: (**a**) angular dense greenish transparent glass with a median content of $SiO_2$ of 76.37% in the sample of 323–325 cm, (**b**) rectangular light semi-transparent dense foliated shard with a SiO2 content of 75.57–76.88% in the sample of 328–3330 cm, (**c**) and (**d**) general view and fragment, respectively, of vesicular, greenish, semi-transparent glass ($SiO_2$ content of 70.38%) in the sample of 334–336 cm, with layered andesitic insertions ($SiO_2$ content of 55.90%) and titanomagnetite crystals and fragments of diatom frustules. Yellow-outlined rectangles show points of measurements.

Pisolites were identified as rounded/semi-rounded grains and fragmentary grains, light (almost white), massive, and occasionally porous and were composed of persilicic and andesitic ash with inclusions of the fragmented diatom frustules, titanomagnetite, and quartz (Figure 6b–d).

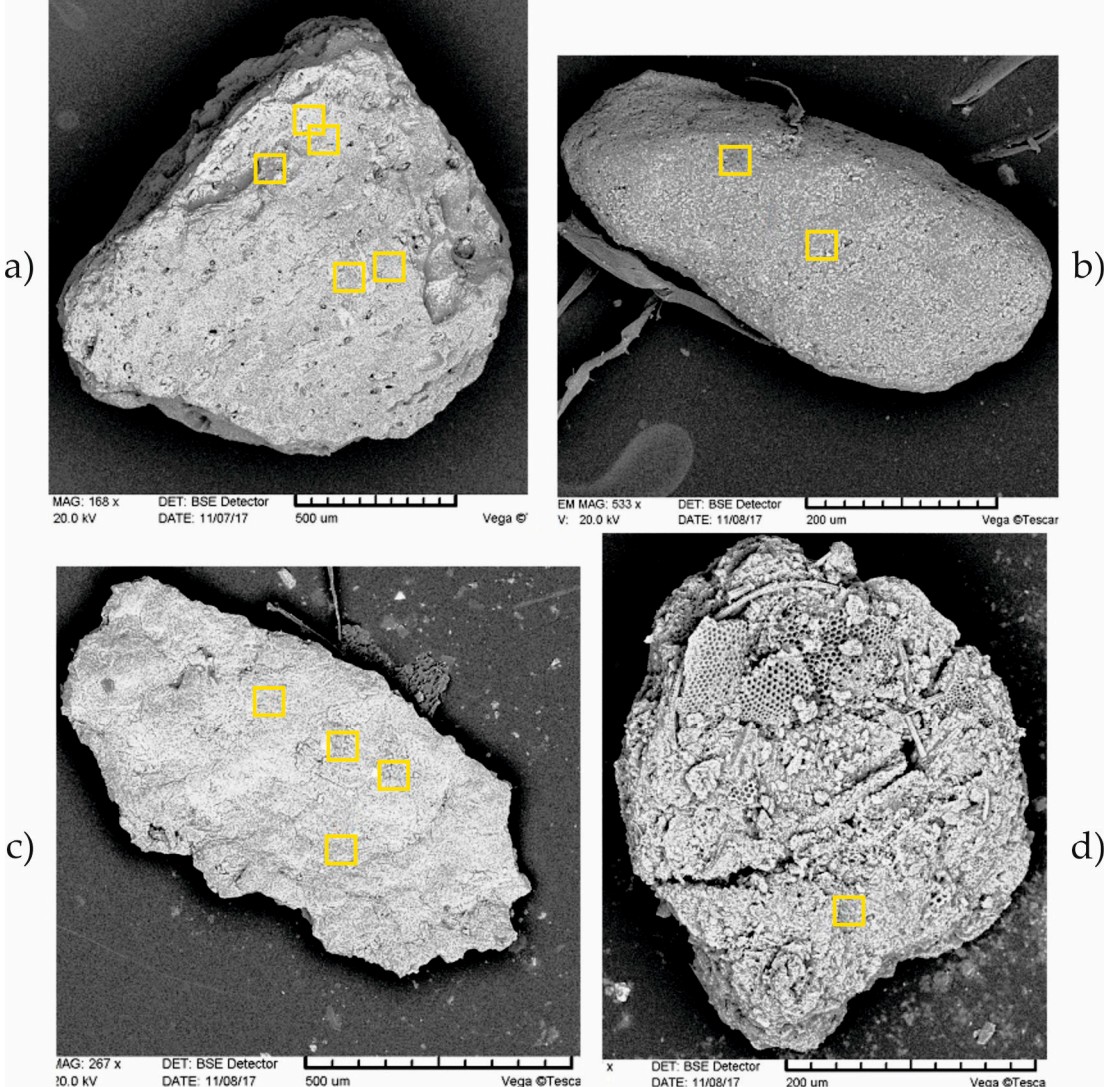

**Figure 6.** Scanning electron microphotographs of the andesitic ash shards and pisolites from the core unit of 323–340 cm: (**a**) rounded fine-pored black heterogeneous andesitic shard with a $SiO_2$ content of 53.30–53.66% in the sample of 323–325 cm; inclusions are the sheet-like fragments with a higher $SiO_2$ content of 55.50–56.05% and small ilmenite crystals; (**b**) light pisolite which is composed of the andesitic basalt ash dust with a $SiO_2$ content of 62.03–65.98% in the sample of 323–325 cm; inclusions are small titanomagnetite crystals; (**c**) unconsolidated grain of light color which is composed of the andesitic basalt ash dust in the sample of 328–330 cm; inclusions are small ilmenite and titanomagnetite crystals; (**d**) white pisolite which is composed of the diatom frustules fragments and volcanic dust in the sample of 328–330 cm. Yellow-outlined rectangles show points of measurements.

Pisolites were found as semi-rounded/non-rounded grains, black, and composed of the andesitic and mafic ash with inclusions of the fragmented diatom frustules, quartz, and pyroxene (Figures 6 and 7a,b,d).

Besides, mostly rounded, transparent, colorless or sometimes rose or yellow-brown (ferruginized) quartz was found.

What are the pisolites in our samples?

Part of the ash material consists of the rounded and semi-rounded aggregates of the very thin, usually slightly cemented ash particles of a size of ≤50 μm having the various components with a sometimes substantial admixture of the fragmented diatom frustules, sponge spicules, and other microfossils. We refer to them as pisolites or ash shatters. They are aggregates of thin volcanic ash. They can form during the penetration of raindrops within ash clouds and during the vapor condensation

on the ash particles in the eruptive clouds [19–22]. According to the classification of the volcaniclastic material in [23], particles <0.1 mm are fine-grained ash dust. This material appears at andesite volcanic eruptions, and the area of its dispersal is unlimited. It can be contaminated with terrestrial particles (e.g., minerals, fresh-water diatoms) during the eolian transport and with marine particles (e.g., microfossils) during the sedimentation in the ocean.

We recognized two types of pisolites in our samples: (1) semi-rounded, black, dense, sometimes pumiceous aggregates of the mixed composition and (2) rounded, white, loose aggregates of the intermediate composition (Figures 6 and 7).

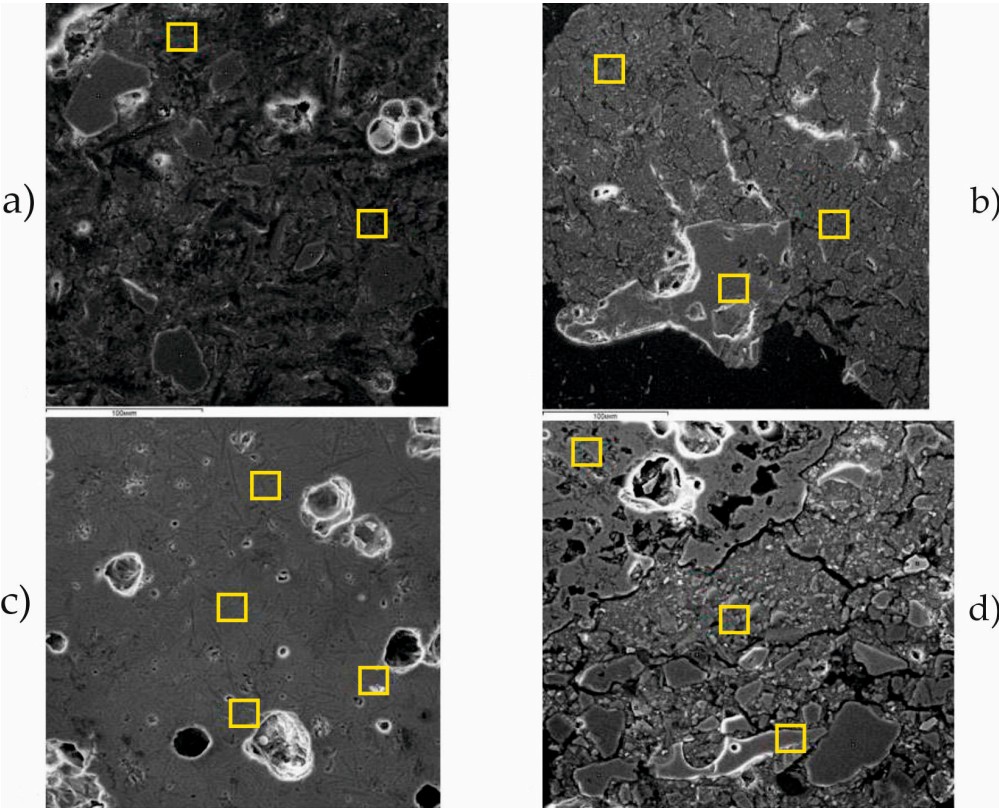

**Figure 7.** Scanning electron microphotographs of the basalts and pisolites of the andesitic and mafic composition on the polished thin sections from the core unit of 323–340 cm: (**a**) Irregularly elongated non-rounded black grain composed of mainly andesitic ($SiO_2$ content of 52.39–58.97%) and sometimes persilicic ($SiO_2$ content of 64.94%) dust in the sample of 328–330 cm. Inclusions are more massive mafic shards, quartz, and microfossil fragments. (**b**) Semi-rounded black shard composed of andesitic ash dust with a mean $SiO_2$ content of 57.5% in the sample of 328–330 cm. Inclusions are occasionally more massive basalt shards with a $SiO_2$ content of 49.22%. (**c**) Microlithic basalt glass with a mean $SiO_2$ content of 50.31% in the sample of 328–330 cm. Microlites are intermediate/mafic plagioclase. (**d**) Semi-rounded black shard composed of andesitic ash dust with a $SiO_2$ content of 54.83% in the sample of 338–340 cm. Inclusions are occasionally larger basalt fragments with a mean $SiO_2$ content of 50.52% and some possible Fe oxides. Yellow-outlined rectangles show points of measurements.

*3.2. Distribution of the Eruptive Material in the Studied Sediment Samples*

3.2.1. The Core Unit of 323–340 cm with an Age of 12,170–12,840 Years

Here, we found abundant ash particles and pisolites, together with foraminiferal shells, from 0.1–0.5 to 2–2.5 mm, sometimes up to 6.5 mm in size; the regular size of the ash shards is between 0.2 and 0.5 mm (Figure 3). Black fragments (basalts, andesitic-basalts, and pisolites of the intermediate and mafic composition) prevail in the upper and lower parts of the unit in the samples of 323–325

and 334–336 cm, respectively, with a content of 30–40% of the whole sediment fraction (Figure 3). The largest fragments of >1–5 mm in size are typical for the upper part of the unit.

The content and size of the black eruptive fragments decrease sharply in the sample of 328–330 cm; they comprised 5–10% of the sediment fraction, and their size was typically <0.5–1 mm. In this sample, the light pisolites of 0.25–0.5 mm in size, composed of persilicic and andesitic ash with quartz, prevailed, reaching 50–70% of the sediment fraction.

The content of the volcanic particles in the lowermost part of the unit, sample 338–340 cm, decreased significantly to 8–10% of the sediment fraction. They are present mostly in the form of black volcanic glass with an admixture of the olive-green ash, white pisolites of the persilicic composition, quartz, and rare fragments of the pyroxene and plagioclase. The size of the particles rarely exceeded 1 mm, with a maximum of up to 2.5 mm.

### 3.2.2. The Core Unit of 355–378 cm with an Age of 13,600–14,540 Years

The eruptive material is present in the form of sharply angular, mainly pumiceous, black basalt fragments with white ash dust in the cavities, the semi-transparent bottle-green, sometimes yellowish, mafic glass particles, quartz, feldspar, and sporadic semi-rounded white pisolites composed of persilicic and andesitic ash dust (Figure 8). In the analyzed sediment fraction, fragments of the eruptive rocks and minerals are larger compared to the biogenic particles, being 0.5–1 mm, occasionally 2–4 mm, in size. The eruptive material within the unit is distributed irregularly with the highest amounts of up to 20–25% of the sediment fraction in the middle (sample 360–362 cm) and lower (sample 376–378 cm) parts; its content in other samples stands at 5–10%.

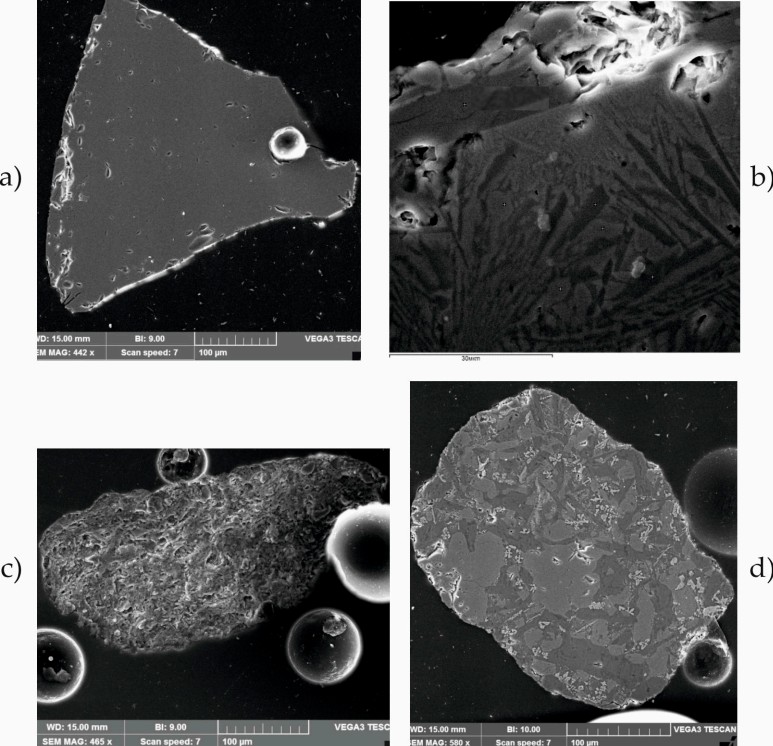

**Figure 8.** Scanning electron microphotographs of the eruptive shards on the polished thin sections from the core unit of 355–378 cm: (**a**) andesitic basalt shard with a $SiO_2$ content of 52.60% in the sample of 355–357 cm, (**b**) microlithic andesitic basalt glass shard with a $SiO_2$ content of 51.68% in the sample of 366-368 cm, inclusions are intermediate/mafic crystals of plagioclase, (**c**) persilicic pisolite with a $SiO_2$ content of 79.1% in the sample of 370–372 cm, inclusions show some crystals of plagioclase, (**d**) dark-grey andesitic pisolite with a $SiO_2$ content of 53.38–56.96% in the sample of 376–378 cm, inclusions are light-grey basalt fragments ($SiO_2$ content of 50.42–51.81%).

*3.3. Chemical Composition of the Volcanic Material in the Studied Samples*

3.3.1. The Core Unit of 323–340 cm with an Age of 12,170–12,840 Years

Sample 323–325 cm

Basalt fragments, which prevail here, have a typical content of $SiO_2$ of 49.05–51.67%, a degraded concentration of $K_2O$ of 0–0.91%, and a high $TiO_2$ amount of 1.49–5.43%. The $TiO_2$ content drops down to 0.68% in some grains with the highest $SiO_2$ concentration. The glass composition on the surface of one shard can vary in some cases from basaltic, with a $SiO_2$ content of 49.10%, to andesitic, with a $SiO_2$ content of 57.15% (Figure 4a). Cavities of the black pumiceous basalt fragments are filled in many cases with volcanic ash dust of the different compositions, from the andesitic one, with a $SiO_2$ content of 55.79%, to the persilicic one, with a $SiO_2$ content of 63.89–65.97% (Figure 4c).

Rhyolites (volcanic persilicic shards) were less numerous than the basaltic fragments. We analyzed the chemical composition of one grain (Figure 5a) where the $SiO_2$ and $K_2O$ contents were high, up to 76.37% and 3.92%, respectively. $TiO_2$ was absent.

Pisolites, which were present here mostly in the form of a white, rounded, loose intermediate variety (Figures 6 and 7), were composed of the ash dust with a $SiO_2$ content from 53.30–53.66% to 58.18% with occasional inclusions of titanomagnetite. Black pisolites had fragments of the mixed mafic ($SiO_2$ content of 42.08–50.76%) to persilicic ($SiO_2$ content of 65.49% on average) composition.

Sample 328–330 cm

Most of the volcanic material was present in the form of pisolites, predominantly of the intermediate-persilicic composition (Figure 6c,d) similar to those in the sample 323–325 cm. Ash dust in pisolites had a $SiO_2$ content of 62.03–65.98%. Inclusions in pisolites were small crystals of titanomagnetite and ilmenite and diatom frustules (Figure 6d).

We also found pisolites of the mixed composition with the andesitic ($SiO_2$ content of 58.97%) and persilicic ($SiO_2$ content of 64.94%) ash dust, larger andesitic basalt particles (mean $SiO_2$ content of 52.39%), occasional quartz inclusions, and a high admixture of the microfossil fragments (mostly diatoms) (Figure 7a). Most black pisolites consisted of the intermediate ash dust with a mean $SiO_2$ content of 57.5% and occasional larger basalt particles (Figure 7b) with a $SiO_2$ content of 49.22%.

Furthermore, there was sporadic mafic with a $SiO_2$ content of 52.04%, intermediate with a $SiO_2$ content of 61.22%, and persilicic ash shards, the latter with a high concentration of $SiO_2$ and $K_2O$, of 75.11% on average and up to 4.88%, respectively. Some mafic shards were microlithic (Figure 7c) and consisted of the basalt glass with a mean $SiO_2$ content of 50.31% and thin mafic plagioclase microliths.

Sample 334–336 cm

The eruptive material consisted predominantly of the ash shards of the mafic (Figure 3d) with a $SiO_2$ content of 50–51%, persilicic with a $SiO_2$ content of 70.49–75.26% and $K_2O$ up to 3.39%, and mixed (Figure 4c,d) composition. Less often, the andesitic basalt shards with a $SiO_2$ content of 52.78% occurred. The persilicic shards are notable here for their elongated shape, presence of the lengthwise scratching, and dense texture. Basalt shards had an irregular shape and were pumiceous (Figures 4 and 5). Shards of the mixed composition were close to the persilicic ones in shape; on their transverse shear surfaces, we could see many small round cavities (bubbles) filled with the fragmented diatom frustules (Figure 5c,d).

Some pisolites consisted of thin andesitic ash dust with a $SiO_2$ content of 59.94%, and abundant diatom fragments were found.

Sample 338–340 cm

Basalt shards with a $SiO_2$ content of 50.17–51.01%, a low $K_2O$ content of <1%, and a high $TiO_2$ concentration of up to 4.44% prevailed in the sample. Andesitic basalt shards with a $SiO_2$ content of

53.46% and sharply angular transparent persilicic glass shards with a $SiO_2$ content of 71.45–71.84% were less numerous. The latter had a relatively high $K_2O$ content of 3.31–3.48%.

Sporadic black, semi-rounded grains of the isometric or oval form of pisolites consisted of andesitic ash dust with a $SiO_2$ content of 52.56–56.30% and occasionally more massive basalt shards (Figure 7d) with a mean $SiO_2$ content of 50.52%.

### 3.3.2. The Core Unit of 355–378 cm with an Age of 13,600–14,540 Years (Figure 8)

We analyzed 16 eruptive grains in the polished thin sections from four samples at 355–357, 360–368, 370–372, and 376–378 cm. Most of them were andesitic basalt shards (Figure 8a) with a $SiO_2$ content of 52.5%; they occurred in all samples of this core unit. The sample 355–357 cm also contained andesitic volcanic shards with a $SiO_2$ content of 53.26 %, microlithic basalt shards with a mean $SiO_2$ content of 51.68% from three analyses, and intermediate-mafic plagioclase crystals (Figure 7b). In the sample of 370–372 cm, we found persilicic glass shards with a $SiO_2$ content of 66.90–67.65% and a very high $K_2O$ concentration of up to 7.93% and persilicic pisolite (Figure 8c) with a $SiO_2$ content of 79.1% and with intruded plagioclase. The sample 376–378 cm had andesitic pisolites with a $SiO_2$ content of 53.38–56.96% or a mean of 55.17% with intruded basalt ($SiO_2$ content of 50.42–51.81% or 51.12% on average) and titanomagnetite fragments (Figure 8d) and basalt shards with $SiO_2$ content of 45.88%.

## 4. Discussion

The diagram of $SiO_2$ versus $K_2O$ in Figure 9 shows that most analyzed volcanic grains from the core unit of 323–340 cm match the ash material from the Icelandic volcano eruptions.

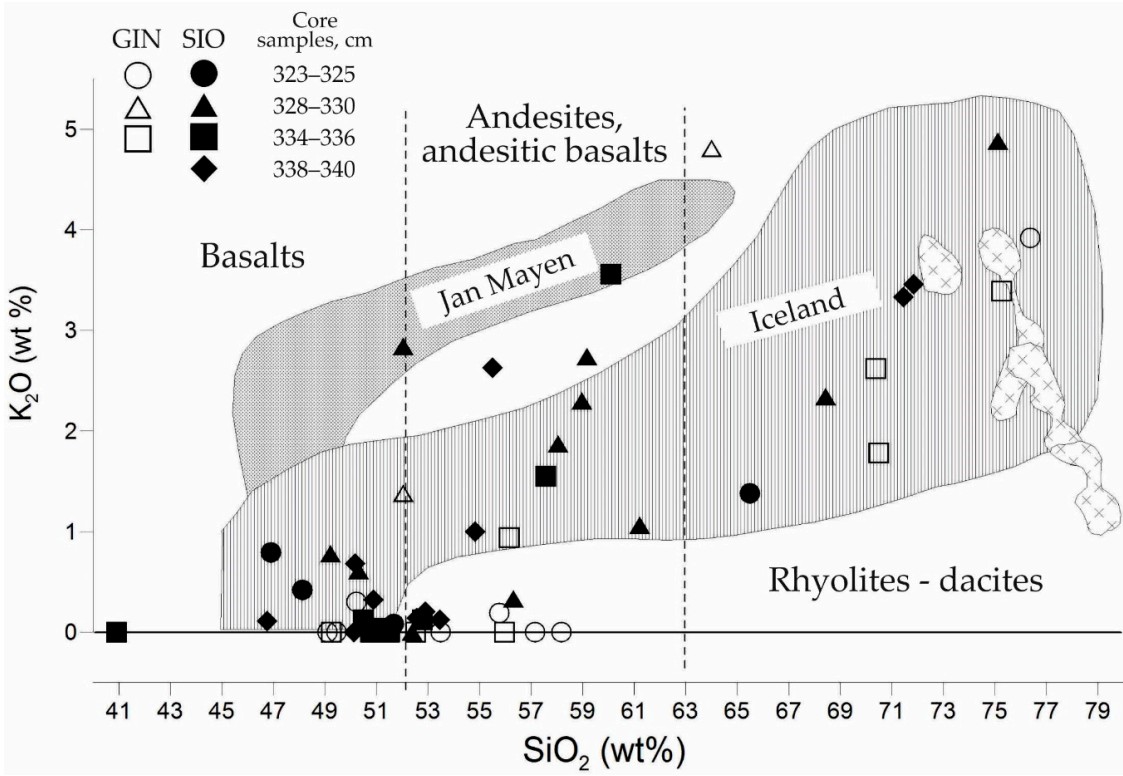

**Figure 9.** Position of the eruptive shards from the ash-bearing core unit of 323–340 cm in the binary $SiO_2/K_2O$ system according to [24]. GIN (Geological Institute) samples present data on the shards in the natural state. SIO (Shirshov Institute of Oceanology) samples show data on the shards in the polished thin sections. Areas of the volcanic material from different sources are according to [25,26].

The sample of 323–325 cm contains basalt shards and pisolites enriched in $TiO_2$ and occasional rhyolites with an elevated $K_2O$ and high FeO concentrations. The bi-plots in Figure 10 present the composition of the presumable Vedde Ash material in the sample of 323–325 cm in comparison with well-known published data. Such ash composition is typical for Vedde Ash whose age in the Greenland ice-core chronology appears to be 12,170 ± 57 years [3,17,18]. A source of the eruptive material for Vedde Ash could be the Katla volcano in Southern Iceland [8,27]. Besides, we found some intermediate ash shards, also as dust in pisolites, with a low $K_2O$ content, but their origin is unclear.

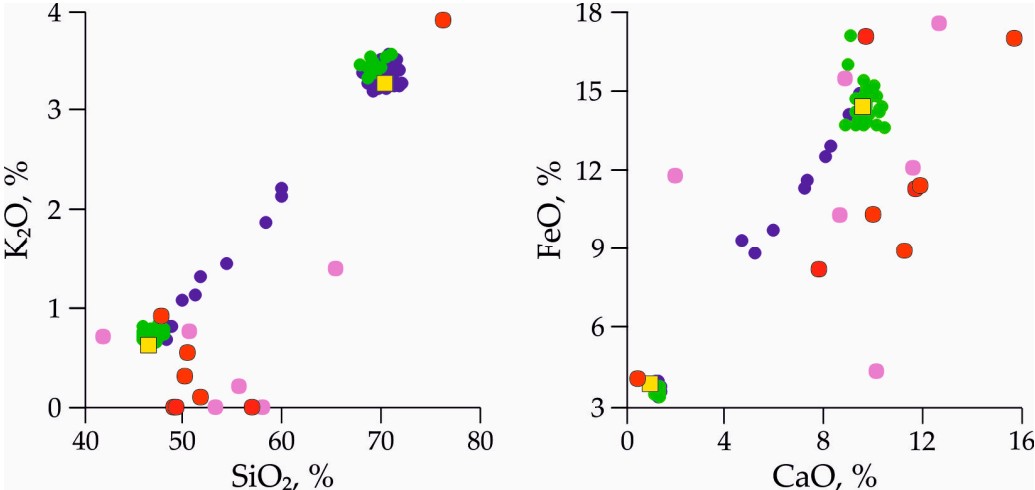

**Figure 10.** Bi-plots of major oxides in the ash material from the core AMK-340 sample of 323–325 cm in comparison to the published data. Black-outlined red circles are the geochemical measurements of the solid shards in the sample of 323–325 cm. Pink circles are the geochemical measurements of the pisolites in the sample of 323–325 cm. Blue circles are data of the Vedde Ash from the lake Kråkenes in western Norway, green circles are data from the lake Loch Ashik in Scotland, and black-outlined yellow squares are data from the Greenland NGRIP ice core [9].

Mangerud et al. [28] and Kvamme et al. [8] summarized findings of Vedde Ash, starting as early as the 1940s, within the Younger Dryas chronozone sediments in the North Atlantic, Nordic Seas, and surrounding European continental areas. Bond et al. [29] and Thornalley et al. [7], as Kvamme et al. [8] before, demonstrated a complicated geochemical composition of the ash beds in the Ash Zone I. Vedde Ash cannot be a simple synonym of the Ash Zone I. Moreover, Lane et al. [9] and Tomlinson et al. [10] concluded that the Icelandic eruptions could have produced ash material comparable to the Vedde Ash before, within and after the Younger Dryas chronozone. Therefore, they claimed an appropriate stratigraphic/chronological control to prove a Vedde-like ash occurrence within the late Younger Dryas time interval. As we presented in the Materials and Methods Section and in Figure 2, the core AMK-340 sample of 323–325 cm, where the possible Vedde Ash is detected, is from the Younger Dryas unit. However, we are aware that this sample does not contain pure Vedde Ash shards but mixed eruptive material originated from possible distal airfall, local sources within the Reykjanes Ridge rift zone, iceberg and sea-ice rafting, reworking during the sedimentation and bioturbation. Unfortunately, we are not able to quantify an exact proportion of the airfall material. An admixture of the local eruptive shards must take place. Ice-rafting influenced the ash accumulation in the area. Ruddiman and Glover [30] described a rhyolitic ash bed in the pre-Holocene sediments in the North Atlantic which can be referred to as the regional Ash Zone I. As the authors suggested, this ash bed must be mostly ice-rafted. Kvamme et al. [20] and Gibbs et al. supposed rather fast Late Pleistocene sea-ice rafting in the North Atlantic ranging from a few years to a few decades, and this is not critical for the time assignment of the core AMK-340 sample of 323–325 cm. According to studies of Ruddiman and Glover [30], the bioturbation can mix the ash through the sediment thickness of dozens of centimeters

at pelagic sedimentation rates of 2.4–6.7 cm $\times$ $10^{-3}$ years. In the core AMK-340, the thickness of the ash-bearing sediments from two core units is 40 cm within the time interval of approximately 12,100–14,500 years B.P. The average sedimentation rate there is approximately 17 cm $\times$ $10^{-3}$ years or much higher compared to that mentioned by Ruddiman and Glover [30]. We can expect a lower than dozens of centimeters sediment mixing depth at the bioturbation within the Younger Dryas, especially if a high activity of the benthic organisms during the cold paleoclimatic environments is not assumed, because the faunal activity (benthic foraminiferas) in the core point started to increase after the Younger Dryas [13]. Studies of the modern sediment mixed depth at the bioturbation in the deep-sea Temperate North Atlantic and Subarctic Nordic Seas exhibited values as high as 3.9 and 6–7 cm [31]. Therefore, we accept possible bottom reworking of the ash-bearing sediments. However, based on the litho- and biostratigraphic information together with the radiocarbon datings, we suppose that the ice-rafting and sediment reworking cannot mask/dispose of the occurrence of the Younger Dryas unit in the core AMK-340. If so, eruptive material from the sample of 323–325 cm can partially have a Vedde Ash origin.

The possible Vedde Ash occurrence in the sediment sample of 323–325 cm allows to re-estimate the timing of the paleoenvironmental changes during the Younger Dryas chronozone (Glacial Stadial 1 in Rasmussen et al. [18]) for the core AMK-340 [13]. Figure 11 presented the age-depth plot for the core AMK-340 based on the radiocarbon datings and corrected the possible Vedde Ash detection with age of 12,170 years [32]. Matul et al. [13] assigned the pre-Holocene warming at the Reykjanes Ridge in the North Atlantic within the Younger Dryas cold chronozone to the time interval of 12,500–12,200 years B.P. or significantly earlier than the beginning of the Holocene (11,700 years B.P.). Now, we can assume that this change occurred 12,200–12,000 years B.P., which was at least 300 years before the Holocene start.

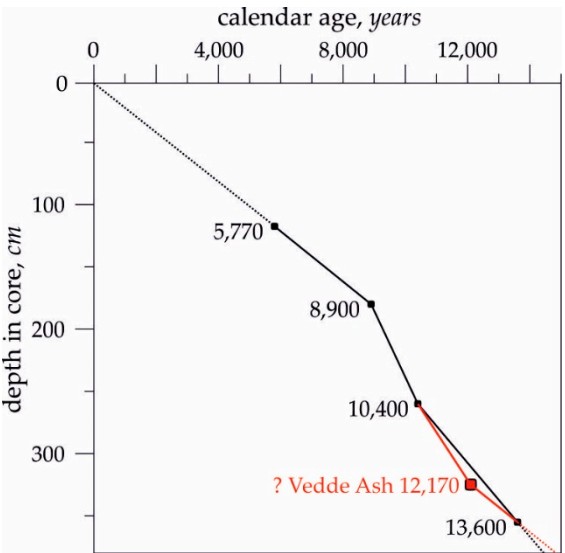

**Figure 11.** Corrected age-depth plot for the Termination I interval of the core AMK-340 (red line) according to the occurrence of Vedde Ash in the sample of 323–325 cm (black-outlined red square). Black squares and black captions on the plot show the radiocarbon datings converted to calendar dates (see also Figure 2).

In the core unit of 328–335 cm with the corrected age estimate of approximately 12,700–12,900 years, the ash is present mainly in the form of andesitic pisolites with the admixture of basalts and rhyolites. Most of them suit the material from the Icelandic volcanoes. However, some basalt fragments with an elevated $K_2O$ in the andesitic pisolites can be related to the Jan Mayen volcanoes (Figure 9). A common occurrence of the marine microfossil (diatom) fragments inside pisolites and ash shard cavities may suggest that the marine transportation and sedimentation could have influenced an accumulation of the volcanic material to a large degree in the area of study. However, we cannot be sure of the

Jan Mayen origin of the above-mentioned andesitic pisolites because Abbot and Davies [3] noted that it is rare to find the Jan Mayen volcanic deposits in the distal areas.

The ash in the core unit of 355–378 cm with an age of 13,600–14,540 years differs from that in the core unit of 323–340 cm with an age of 12,170–12,840 years.

In the samples of 355–368 cm and 376–378 cm with an age of 13,600–14,100 and 14,540 years, respectively, we can see a dominance of the basalt and andesitic basalt shards with a $SiO_2$ content of 52.32–53.13%, FeO content of 9.71–11.13%, MgO content of 6.31–8.69%, and $K_2O$ content of <1%. In terms of the geochemical composition, they are close to the tholeiitic basalts and basalt glass in the rift zone of the Reykjanes Ridge [12] (Figure 12) but differ from them in terms of a higher $SiO_2$ concentration. The $SiO_2$ content is 49.76% and 50.56% on average in the tholeiitic basalts and the basalt shards of the Reykjanes Ridge, respectively. The ash in the core unit of 355–378 cm may be mainly the local eruptive material originated from the volcanism and tectonics in the Reykjanes Ridge rift zone [12] and may have transformed under the influence of the acidic hydrothermal fluids during sedimentation.

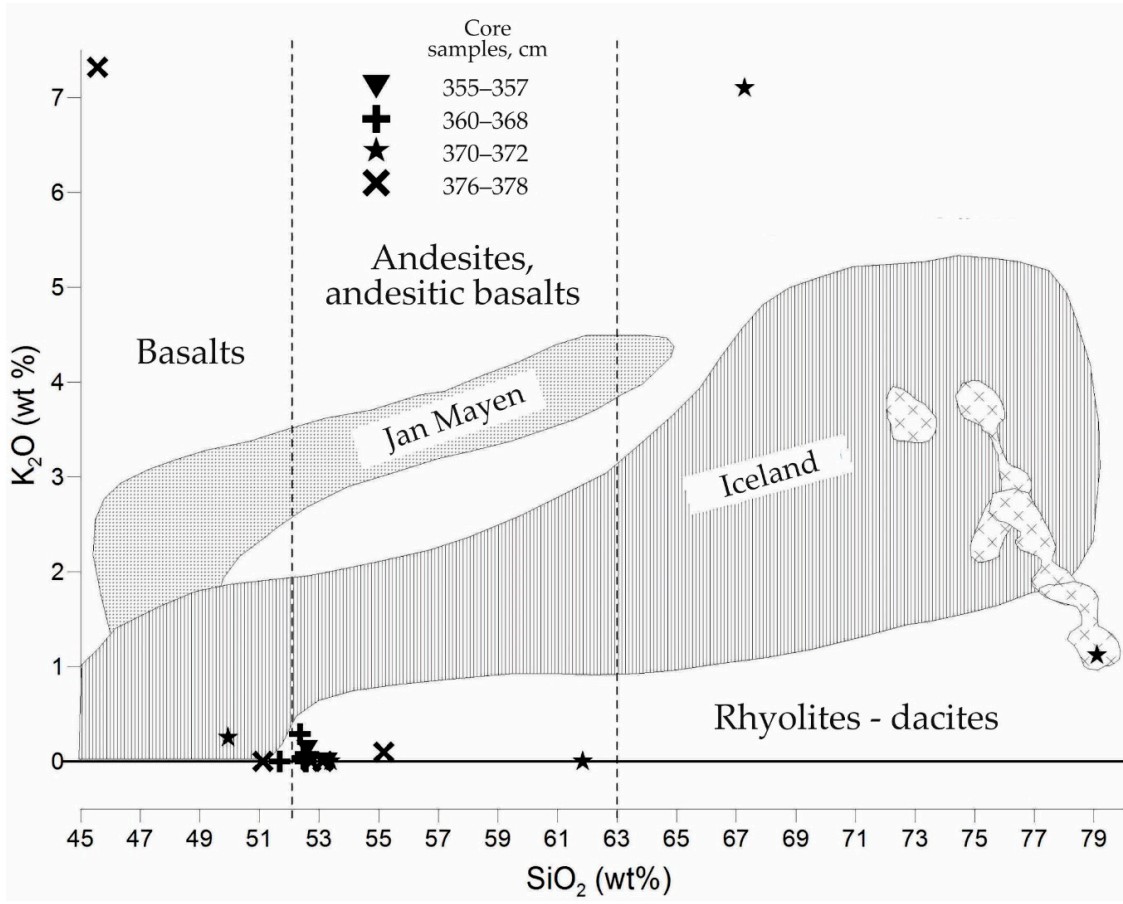

**Figure 12.** Position of the eruptive shards from the ash-bearing core unit of 355–378 cm in the binary $SiO_2/K_2O$ system according to [24]. All data on our samples are from the polished thin sections. Areas of the volcanic material from different sources are according to [25,26].

Other ash materials in the core unit of 355–378 cm are (1) mafic shards with a decreased $SiO_2$ content of 45.88% in the lowermost sample of 376–378 cm and (2) persilicic glass with a $SiO_2$ content of 67.65% and a high $K_2O$ concentration in the sample of 370–372 cm; the sources of both are unclear. Persilicic pisolites in the sample of 370–372 cm and mafic pisolites with a $SiO_2$ content of 51.12% in the sample of 376–378 cm can have a possible origin from the Icelandic volcanoes, but the source of the andesitic pisolites with a $SiO_2$ content of 55.17% in the sample of 376–378 cm is not defined and could be more distal.

## 5. Conclusions

Two sediment units of the core AMK-340, Reykjanes Ridge, North Atlantic, contain a significant amount of volcanic ash. They can be related to the Ash Zone I in the North Atlantic Late Quaternary sediments. Accumulation of the ash-bearing sediments in the studied core samples occurred within the time intervals of 12,170–12,840 and 13,600–14,540 years B.P. or the Younger Dryas cold chronozone and Bølling–Allerød warm chronozone, respectively.

The ash in the core AMK-340 within the Younger Dryas unit is present mostly in the form mafic and persilicic material originated from the Icelandic volcanoes, and ice-rafted to the point of the study. In one sediment sample, we could detect shards of possible bimodal Vedde Ash. If it was accepted that the core sample of 323–325 cm contains Vedde Ash, the age model of the core AMK-340 and timing of the previously reconstructed paleoceanographic changes for the Termination I time interval can be specified. However, in any case, significant warming in the area could have occurred as early as 300 years before the end of the conventional Younger Dryas cold chronozone.

The ash in the core AMK-340 within the Bølling–Allerød unit is present mostly in the form of mafic shards which are close to the basalts and basalt glass of the rift zone in the Reykjanes Ridge, i.e., have presumably a local origin.

In some samples of both units, we found aggregated pisolites of andesitic ash dust with inclusions of persilicic/mafic particles and fragments of marine diatom frustules. The origin of these pisolites is unclear.

**Supplementary Materials:** The following are available online at http://www.mdpi.com/2076-3263/9/9/379/s1, Table S1: Major oxides (weight %) in the analyzed ash shards from the sediment fraction of >100 μm in samples of the core AMK-340.

**Author Contributions:** Conceptualization, A.M.; methodology, I.F.G.; investigation, I.F.G., T.A.K., A.I.M.; writing—original draft preparation, A.M., I.F.G.; writing—review and editing, A.M., I.F.G.; visualization, N.V.L.

**Funding:** Funds for this research were acquired from the Ministry of Education and Science of the Russian Federation, Project No. 0149-2019-0007 for the Shirshov Institute of Oceanology, and additional funding was obtained from Project No. 0135-2019-0050 for the Geological Institute (IFG and AIM).

**Acknowledgments:** We thank two anonymous reviewers for the constructive criticism and valuable suggestions which helped to improve our paper significantly. The authors are grateful to N.V. Gorkova, V.V. Mikheev, A.G. Boev, G.Kh. Kazarina, and A.V. Tikhonova for their valuable consultations and technical support.

**Conflicts of Interest:** The authors declare no conflict of interest. The funders had no role in the design of the study; in the collection, analyses, or interpretation of data; in the writing of the manuscript, or in the decision to publish the results.

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
