# Peer review of "The Distal and Local Volcanic Ash in the Late Pleistocene Sediments of the Termination I Interval at the Reykjanes Ridge, North Atlantic, Based on the Study of the Core AMK-340"

_geosciences, doi:10.3390/geosciences9090379_

Round 1
Reviewer 1 Report
Review of Matul et al
Overview
This is a potentially interesting paper, reporting some unusual chemistries in Ash that is likely to be Ash Zone 1. However, I have a number of concerns, below, relating to the reporting of the data that mean this is not ready for publication. In particular, as the main findings are reported to be the Vedde and some unknown tephra chemistry, full reporting of the data, including secondary standards is essential. At the moment I am not convinced that the vedde has been found, and the unusual chemistry might be a product of either analytical issues, or a mix of mineral and glass during individual analyses of shards. There are also number of other points raised below. Without reporting of the underlying data and a clear discussion of data reliability (see points about analyses of non-polished samples and energy dispersive data) this paper is not ready for publication.
Intro and Methods
Line 54 please show a full lithostratigraphy for the relevant section of the core (picture or log)
Line 65-70 - This is only a partial cryptotephra study and is going to pick up a significant component of IRD derived tephra shards, see e.g. the work of Griggs et al trying to correlate marine records in this region using tephra. There needs to be a much clearer strategy in working out aitfall from ice rafted material, as there can be a significant time lag in the latter. It is also not clear from this section if there is a background of ~10% shards of greater than 100 micron through the whole core. Ciritally you need to explain the details of the microprobe operating conditions, what secondary standards were used, how reliably these are repeated in analyses. It is also unlikely that data from unpolished shards is going to be reliable. It is also often very difficult to get reliable data from Energy dispersive analyses alone, are data raw or normalised. Also the shard pictures in the results suggest mineral inclusions and large shards and thus the beam diameter and use of backscatter imaging is important in producing reliable data. Please explain in more detail how the chemistry was produced.
Line 84-92 Far more detail is needed on the radiocarbon ages, please show a table of radiocarbon dates, an explanation of reservoir correction, the detail of the age modelling process, lab numbers d13C and so on. You also need to prove the presence of the Vedde ash before you use it in the age model.
Results and Discussion
There is clearly some reworking of material in the core, hence the mix of mineral and other material in the rounded clasts and this highlights the clear need for a much better developed strategy to discriminate between primary airfall and reworked material. Some thin section analyses would help a lot here.
The presentation and interpretation of the data is not sufficient for an international publication. A table of raw data needs to be available at least for the review stage and a link to where the data is to be accessed afterwards, preferably hosted as supplementary material on the journal page. It is not possible as a reviewer to examine the quality or interpretation of the data as it is currently presented.
A figure is also needed for 328-8125px showing the composition of the Vedde Ash for peer reviewed sources (for example Tomlinson et al or Lane et al) and the data in this paper, to show that the Vedde has been found. This also needs to be clearly placed in the context of the age model as it is well known that Katla basalts are common and the rhyolite phase of the Vedde is not chemically unique, and only if you are sure that the core is mid Younger Dryas in age can you be sure you have the Vedde (see Lane et al - is the Vedde one of a kind).
Author Response
Dear Reviewer #1,
on behalf of my co-authors, I’d like to thank you for your thorough and very constructive review of our manuscript which helped to improve it significantly. I understand your concerns, and tried to do my best to provide the appropriate answers.
Overall, the manuscript was changed significantly: now 19 pages instead of 14, 12 figures instead of 9, 1 table was added in the text, 1 table with the analytical data was added as a supplement, and more references were added. Many paragraphs were rewritten/extended, new parts of the text were added. Following your recommendations, we mentioned works of Griggs et al., Tomlinson et al., Lane et al., and some other papers. We also tried to be more cautious with our conclusions about the Vedde Ash detection. English was checked and corrected. You can track all the changes reading the *.docx manuscript.
---------------------------------------------------
Comments and Suggestions for Authors
Overview
This is a potentially interesting paper, reporting some unusual chemistries in Ash that is likely to be Ash Zone 1. However, I have a number of concerns, below, relating to the reporting of the data that mean this is not ready for publication. In particular, as the main findings are reported to be the Vedde and some unknown tephra chemistry, full reporting of the data, including secondary standards is essential. At the moment I am not convinced that the vedde has been found, and the unusual chemistry might be a product of either analytical issues, or a mix of mineral and glass during individual analyses of shards. There are also number of other points raised below. Without reporting of the underlying data and a clear discussion of data reliability (see points about analyses of non-polished samples and energy dispersive data) this paper is not ready for publication.
Table *.xlsx with the analytical data on major oxides is now in the Supplement.
Intro and Methods
Line 54 please show a full lithostratigraphy for the relevant section of the core (picture or log)
new Figure 2 (see line 101) is added giving main litho-and biostratigraphic information on the core.
Line 65-70 - This is only a partial cryptotephra study and is going to pick up a significant component of IRD derived tephra shards, see e.g. the work of Griggs et al trying to correlate marine records in this region using tephra. There needs to be a much clearer strategy in working out aitfall from ice rafted material, as there can be a significant time lag in the latter. It is also not clear from this section if there is a background of ~10% shards of greater than 100 micron through the whole core. Ciritally you need to explain the details of the microprobe operating conditions, what secondary standards were used, how reliably these are repeated in analyses. It is also unlikely that data from unpolished shards is going to be reliable. It is also often very difficult to get reliable data from Energy dispersive analyses alone, are data raw or normalised. Also the shard pictures in the results suggest mineral inclusions and large shards and thus the beam diameter and use of backscatter imaging is important in producing reliable data. Please explain in more detail how the chemistry was produced.
In Introduction (lines 60-63), and then in Discussion (lines 737-761), we described the environmental processes (with references) which could affect a deposition of the ash material in the core sediments. In Material and Methods, there is now a paragraph (lines 214-232) presenting the analytical instruments; also, Table 1 (line 146) gives information on the number of the analyzed shards and measurements.
Line 84-92 Far more detail is needed on the radiocarbon ages, please show a table of radiocarbon dates, an explanation of reservoir correction, the detail of the age modelling process, lab numbers d13C and so on. You also need to prove the presence of the Vedde ash before you use it in the age model.
New Figure 2 and extended paragraph in Material and Methods (lines 114-133) provide such information.
Results and Discussion
There is clearly some reworking of material in the core, hence the mix of mineral and other material in the rounded clasts and this highlights the clear need for a much better developed strategy to discriminate between primary airfall and reworked material. Some thin section analyses would help a lot here.
As can be seen from Table 1, from the total 141 measurements, 112 are made on the polished thin sections. Table S1 (supplemented file) presents data separately on the natural shards and polished thin sections, the same on the solid grains and pisolites.
The presentation and interpretation of the data is not sufficient for an international publication. A table of raw data needs to be available at least for the review stage and a link to where the data is to be accessed afterwards, preferably hosted as supplementary material on the journal page. It is not possible as a reviewer to examine the quality or interpretation of the data as it is currently presented.
Table S1 is now in the Supplement.
A figure is also needed for 328-8125px showing the composition of the Vedde Ash for peer reviewed sources (for example Tomlinson et al or Lane et al) and the data in this paper, to show that the Vedde has been found. This also needs to be clearly placed in the context of the age model as it is well known that Katla basalts are common and the rhyolite phase of the Vedde is not chemically unique, and only if you are sure that the core is mid Younger Dryas in age can you be sure you have the Vedde (see Lane et al - is the Vedde one of a kind).
Figure 10 (line 720) presents two bi-plots: our data on the sample 323-325 cm (possible Vedde Ash level) vs data from Lane et al. Description of the age model (Material and Methods in lines 99-133 + Figure 11 in line 771) allow us to suggest that this is the sample within the Younger Dryas. Of course, we clearly see that the ash material in this sample is not pure Vedde Ash, and other samples also contain the Icelandic shards (and we expressed this in the text, e.g., Discussion lines 776-779). However, for us it was more important to demonstrate that the Termination I interval in our core has two very distinct ash-bearing units: 1) with predominantly local eruptive material within the Boelling-Alleroed chronozone, and 2) with mixed local and distal (? Icelandic) material within the Younger Dryas. In Conclusions we made more cautious sentences (lines 841-845) about the Vedde Ash detection in our core.
Thank you very much again!
We are ready to meet your possible additional comments/questions on this revised manuscript.
Matul
Reviewer 2 Report
This paper uses a marine core taken from the Reykjanes ridge and attempts to identify the source of the volcanic ash (tephra) which is contained within the core. In particular they attempt to differentiate some of the source volcanic systems for the tephra found in Ash Zone 1, and ascribe an ash rich unit at 323-325 cm depth as the Vedde Ash. This identification is then used to amend the age model of the core and to date the palaeoclimatic changes in this region. 65 shards were selected for geochemical analysis in total.
Its hard to judge based on the information here how significant this work is for our understanding if the wider context of environmental change during this time period. It would be good if the authors spend a little more time elaborating what the findings in this core might mean for our understanding of change at this location (e.g ocean circulation, extent of ice rafting). This aspect could be developed more in the conclusion section.
The work provides a contribution mainly through its putative identification of the vedde ash, which adds to our understanding of the distribution of the vedde ash but also could alter the age model for this core which was previously based solely on radiocarbon dates. This might alter the interpretation of the wider palaeoceanographic changes at this location.
The method and approach appears to be broadly sound. However, more information on the method for the tephra major element geochemical analysis needs to be provided. Particularly lacking is information on the instrument conditions for the microprobe analysis – it is standard practice to include this information. I also strongly recommend that the oxide totals are included in a supplementary information section so the dataset is available for comparison – and this should make clearer the sample depth and the tephra analysed. Ideally, the dataset should also be uploaded to a data repository as well so other researchers can access it easily – e.g. tephrabase.org
One of the major contributions claimed by this paper is the identification of the vedde ash in their core. I would like to see a more robust approach to the identification of this tephra layer in the paper as the way in which the evidence is presented its not clear enough. How many shards were analysed in 323-325 cm? Ideally you would have a separate bi-plot for this, with not just the zones for different volcanic sources, but actual other published data from the vedde ash plotted on it as well. This would make it easier to assess the strength of this identification.
An additional area which needs developing is the age model – which is not shown in the paper (although it is referenced in other papers). The revised age model with the Vedde ash identification should be plotted in a diagram and included in the paper, since it forms a key part of the main findings of the paper.
Some further minor comments:
There are lots of photos, and its not clear enough to me why they all merit inclusion in the main text – suggestion to move some to SI so that there is more room for summary diagram.
It is noted that the size of the tephra shards found were all large – the paper would benefit from some further comment on the size of tephra shards found and analysed, if that information was collected.
Oxide weights could probably be presented to just one significant figure in the text.
Note of typographical or presentation issues
I think the paper would benefit from further editing from a typographical perspective – there are numerous minor errors. On the whole it is clear and well written and easy enough to follow.
Line 45 – unnecessary an before ‘eruptive’
Line 129-130 – not clear what this sentence means
Line 131 – should sentence be title? (fragment)
Line 155 – remove the ? – change to ‘possible Fe oxides’
Line 253 – 1940th(typo)
Line 274 – volcanoe (sp)
Author Response
Dear Reviewer #2,
I appreciate very much your constructive comments and suggestions to our manuscript! They helped to revise the manuscript in a large extent. Together with co-authors, I tried to answer all your questions.
Overall, the manuscript was changed significantly: now 19 pages instead of 14, 12 figures instead of 9, 1 table was added in the text, 1 table with the analytical data was added as a supplement, and more references were added. Many paragraphs were rewritten/extended, new parts of the text were added. We mentioned the tephra works of Griggs et al., Tomlinson et al., Lane et al., and some other papers. We also tried to be more cautious with our conclusions about the Vedde Ash detection. English was checked and corrected. You can track all the changes reading the *.docx manuscript.
Comments and Suggestions for Authors
This paper uses a marine core taken from the Reykjanes ridge and attempts to identify the source of the volcanic ash (tephra) which is contained within the core. In particular they attempt to differentiate some of the source volcanic systems for the tephra found in Ash Zone 1, and ascribe an ash rich unit at 323-325 cm depth as the Vedde Ash. This identification is then used to amend the age model of the core and to date the palaeoclimatic changes in this region. 65 shards were selected for geochemical analysis in total.
During the micropaleontological examination of the sediment fractions of >100 µm, we realized that some samples contain a lot of the ash. It was for the core unit of the Termination I. And it seemed to be that a composition of the volcanic material in these samples was different. Geochemical analyses demonstrated two more or less large sediment units: one with a prevalence of the local material from the rift zone, and another with mixed local and distal (Icelandic) material. The latter – within the Younger Dryas. So that, we wanted to report mostly this finding. Possible detection of the Vedde material was also fine as we could get one more chronological marker.
Its hard to judge based on the information here how significant this work is for our understanding if the wider context of environmental change during this time period. It would be good if the authors spend a little more time elaborating what the findings in this core might mean for our understanding of change at this location (e.g ocean circulation, extent of ice rafting). This aspect could be developed more in the conclusion section.
Paleoceanographic implication from the study of the core are presented in the publication of Matul et al 2018 – see # 13 in the Reference list of this manuscript. I’d like just repeat that our micropaleontological and oxygen-isotopic study of the core exhibited local earlier start of the final warming within the last deglaciation on the subsurface depths: some centuries before the Younger Dryas end.
The work provides a contribution mainly through its putative identification of the vedde ash, which adds to our understanding of the distribution of the vedde ash but also could alter the age model for this core which was previously based solely on radiocarbon dates. This might alter the interpretation of the wider palaeoceanographic changes at this location.
Possible detection of the Vedde level can specify the age model but does not change significantly our paleoceanographic conclusions (just some shift of timing) – please, see my comment to the previous paragraph.
The method and approach appears to be broadly sound. However, more information on the method for the tephra major element geochemical analysis needs to be provided. Particularly lacking is information on the instrument conditions for the microprobe analysis – it is standard practice to include this information. I also strongly recommend that the oxide totals are included in a supplementary information section so the dataset is available for comparison – and this should make clearer the sample depth and the tephra analysed. Ideally, the dataset should also be uploaded to a data repository as well so other researchers can access it easily – e.g. tephrabase.org
In Material and Methods, there is now a paragraph (lines 214-232) presenting the analytical instruments. Also, Table 1 (line 146) gives information on the number of the analyzed shards and measurements. Besides, Table S1 is now in Supplement presenting the analytical data on major oxides from our samples.
One of the major contributions claimed by this paper is the identification of the vedde ash in their core. I would like to see a more robust approach to the identification of this tephra layer in the paper as the way in which the evidence is presented its not clear enough. How many shards were analysed in 323-325 cm? Ideally you would have a separate bi-plot for this, with not just the zones for different volcanic sources, but actual other published data from the vedde ash plotted on it as well. This would make it easier to assess the strength of this identification.
As Reviewer # 1 has the same concern, I repeat my answer:
Figure 10 (line 720) presents two bi-plots: our data on the sample 323-325 cm (possible Vedde Ash level) vs data from Lane et al. Description of the age model (Material and Methods in lines 99-133 + Figure 11 in line 771) allow us to suggest that this is the sample within the Younger Dryas. Of course, we clearly see that the ash material in this sample is not pure Vedde Ash, and other samples also contain the Icelandic shards (and we expressed this in the text, e.g., Discussion lines 776-779). However, for us it was more important to demonstrate that the Termination I interval in our core has two very distinct ash-bearing units: 1) with predominantly local eruptive material within the Boelling-Alleroed chronozone, and 2) with mixed local and distal (? Icelandic) material within the Younger Dryas. In Conclusions we made more cautious sentences (lines 841-845) about the Vedde Ash detection in our core.
An additional area which needs developing is the age model – which is not shown in the paper (although it is referenced in other papers). The revised age model with the Vedde ash identification should be plotted in a diagram and included in the paper, since it forms a key part of the main findings of the paper.
New Figure 2 and extended paragraph in Material and Methods (lines 114-133) provide such information. Figure 11 presents a modified age-depth plot according to the possible detection of the Vedde material.
Some further minor comments:
There are lots of photos, and its not clear enough to me why they all merit inclusion in the main text – suggestion to move some to SI so that there is more room for summary diagram.
Figures with photos were modified: points of the measurements for each grain were marked. To our mind, this can add to better presentation of results. If you are not completely negative, we would like to leave photos within the text.
It is noted that the size of the tephra shards found were all large – the paper would benefit from some further comment on the size of tephra shards found and analysed, if that information was collected.
Some text about was added to Material and Methods, and Results (subchapters 3.1.1-3.1.2).
Oxide weights could probably be presented to just one significant figure in the text.
Table S1 with major oxide weights was added to Supplement. Some data is presented on the bi-plots in Figure 10.
Note of typographical or presentation issues
Ok, checked.
I think the paper would benefit from further editing from a typographical perspective – there are numerous minor errors. On the whole it is clear and well written and easy enough to follow.
Ok, checked.
Line 45 – unnecessary an before ‘eruptive’
Line 129-130 – not clear what this sentence means
Line 131 – should sentence be title? (fragment)
Line 155 – remove the ? – change to ‘possible Fe oxides’
Line 253 – 1940th(typo)
Line 274 – volcanoe (sp)
Ok, we made all corrections.
Thank you very much again!
A. Matul